# Glycomacropeptide Safety and Its Effect on Gut Microbiota in Patients with Phenylketonuria: A Pilot Study

**DOI:** 10.3390/nu14091883

**Published:** 2022-04-29

**Authors:** Chiara Montanari, Camilla Ceccarani, Antonio Corsello, Juri Zuvadelli, Emerenziana Ottaviano, Michele Dei Cas, Giuseppe Banderali, Gianvincenzo Zuccotti, Elisa Borghi, Elvira Verduci

**Affiliations:** 1Department of Pediatrics, Vittore Buzzi Children’s Hospital, University of Milan, 20154 Milan, Italy; chiara.montanari@asst-fbf-sacco.it (C.M.); antonio.corsello@unimi.it (A.C.); gianvincenzo.zuccotti@unimi.it (G.Z.); elvira.verduci@unimi.it (E.V.); 2Institute of Biomedical Technologies, National Research Council, 20090 Segrate, Italy; camilla.ceccarani@itb.cnr.it; 3Clinical Department of Pediatrics, ASST Santi Paolo e Carlo, San Paolo Hospital, University of Milan, 20142 Milan, Italy; juri.zuvadelli@asst-santipaolocarlo.it (J.Z.); giuseppe.banderalii@asst-santipaolocarlo.it (G.B.); 4Department of Health Science, University of Milan, 20142 Milan, Italy; emerenziana.ottaviano@unimi.it (E.O.); michele.deicas@unimi.it (M.D.C.)

**Keywords:** gut microbiota, phenylketonuria, glycomacropeptide, GMP, diet, calcium homeostasis, nutritional therapy, vitamin D, *Agathobacter*, *Subdoligranulum*

## Abstract

Glycomacropeptide (GMP) represents a good alternative protein source in Phenylketonuria (PKU). In a mouse model, it has been suggested to exert a prebiotic role on beneficial gut bacteria. In this study, we performed the 16S rRNA sequencing to evaluate the effect of 6 months of GMP supplementation on the gut microbiota of nine PKU patients, comparing their bacterial composition and clinical parameters before and after the intervention. GMP seems to be safe from both the microbiological and the clinical point of view. Indeed, we did not observe dramatic changes in the gut microbiota but a specific prebiotic effect on the butyrate-producer *Agathobacter* spp. and, to a lesser extent, of *Subdoligranulum*. Clinically, GMP intake did not show a significant impact on both metabolic control, as phenylalanine values were kept below the age target and nutritional parameters. On the other hand, an amelioration of calcium phosphate homeostasis was observed, with an increase in plasmatic vitamin D and a decrease in alkaline phosphatase. Our results suggest GMP as a safe alternative in the PKU diet and its possible prebiotic role on specific taxa without causing dramatic changes in the commensal microbiota.

## 1. Introduction

Phenylketonuria (PKU) is an inherited metabolic disorder caused by the mutation of the phenylalanine hydroxylase enzyme, which converts phenylalanine (Phe) into tyrosine. The impaired activity of this enzyme leads to Phe accumulation in the blood, which is toxic to the brain. An early diagnosis and treatment can prevent neurodevelopmental damage. Currently, the dietary approach represents the mainstay of PKU therapy. According to European guidelines for PKU management, a special “low Phe intake” diet should be started as early as possible [1,2], with the aim of maintaining Phe concentrations within safe ranges. The neonatal period corresponds to a well-recognized crucial step in microbiota acquisition and maturation [3], on which environmental factors may have a profound impact. 

The gut microbiota, a complex microbial ecosystem that resides largely in the distal bowel, represents the most studied microbial population thanks to the unique functions it performs, contributing to maintaining human health [4]. The individual host-associated gut microbiota seems to be the result of a genetic substrate on which external factors play their role, influencing the final composition and consequent functions. Among these, in addition to antibiotics, infectious agents, and lifestyle, diet seems to be one of the most important shaping factors [5], so different dietary patterns are related to characteristic bacteria profiles [6,7]. The bioavailability of specific dietary substrates and their quality can influence the growth of different microbial species and, consequently, their short-chain fatty acids (SCFAs) production [8,9]. In PKU patients, a well-defined dietary treatment with Phe-free-L-amino acid mixtures and low special protein foods provides for a restriction of natural proteins, guaranteeing at the same time an adequate intake of proteins and proper growth. Recent studies analyzing the intestinal microbial profile of PKU patients showed low microbial biodiversity [8,10,11], considered suggestive of intestinal microbiota dysbiosis [12] and the depletion of some beneficial genera. This condition seems to be driven by the quality of dietary nutrients characterizing the PKU diet [13] since daily glycemic index and glycemic load resulted higher in PKU patients compared with mild hyperphenylalaninemia (MHP) subjects on a free diet [8].

Recently, some companies involved in PKU therapeutic feeding started producing specific medical foods based on Glycomacropeptide (GMP). GMP is a protein derived from cheese whey, rich in some specific essential amino acids, principally threonine and isoleucine, and, in its pure form, it is almost free of Phe, tyrosine, and tryptophan (aromatic amino acids). For this reason, commercial GMP formulations are generally supplemented with tyrosine and tryptophan in order to avoid eventual deficiencies, and they could represent a good alternative protein source for PKU patients with better palatability than other amino acids formulas [14]. 

Besides the above-mentioned properties, in a rodent model, the GMP seems to exert a prebiotic role on microbiota since its structure is characterized by extensive glycosylation with sugars (sialic acid, galactosyl, and N-acetylgalactosamine), which are substrates for some specific beneficial bacteria such as Lactobacillus and Bifidobacteria. Moreover, an immunomodulatory influence was reported in mice fed with GMP (reduced plasmatic concentrations of IFN- γ, TNF-α, IL-1β, and IL-2) with an anti-inflammatory role and subsequent positive systemic effects on the health status of the host [14].

In the present study, we evaluated the effect of GMP supplementation on the gut microbial community of PKU patients by comparing their microbial community before and after the intervention. A long-term intervention, i.e., 6 months, was planned in order to evaluate stable GMP-derived changes in the microbial community [15].

## 2. Materials and Methods

### 2.1. Cohort

Patients were enrolled at the Pediatric Department of ASST Santi Paolo e Carlo in Milan. Inclusion criteria were: PKU diagnosis by newborn screening, age ≥ 6 years, annual mean Phe levels within the range (i.e., 120–360 µmol/L) in childhood (<12 years) and 120–600 µmol/L in adolescence and adult age (>12 years), as recommended by the European PKU guidelines. Phe values were calculated as the average between measurements taken in the 6 months before the intervention and the value at the enrolment. Exclusion criteria were: a previous GMP intake, congenital malformations, endocrine disorders, chronic liver diseases, chronic or acute intestinal diseases, treatments with antibiotics, and probiotic/prebiotic in the 3 months preceding the study.

### 2.2. Clinical Data Collection

GMP intervention, aimed at replacing the amino acids with GMP, started after the baseline visit, with an imbrication phase lasting no more than 4 weeks, reaching at least 30% of the amino acid supplementation from GMP formulation at T1.

At bothT0 (baseline) and T1 (6 months of GMP intake), the following analyses were performed: anthropometry, including calculation of body mass index; body composition (air displacement plethysmography), nutrients intake, and Glycemic index (GI); Glycemic load (GL) by a 3–day food diary; hematological and biochemical status; and microbiota characterization. 

Weight, height, and body composition data were collected by metabolic dietitians. Height was measured using an electronic stadiometer (Seca GmbH & Co. Kg., model 242, Hamburg, Germany). Weight and body composition were measured with BodPod GS (COSMED, Concord, CA, USA); the data are expressed as fat mass (FM; kg; %) and free fat mass (FFM, kg; %). 

#### 2.2.1. Nutritional Assessment

For each patient, the dietary intake was recorded by means of a food diary filled out by the patient or caregiver for three consecutive days (two weekdays and one weekend day) at both T0 (baseline) and T1 (6 months of GMP intake). Parents received instructions about the method for weighing and recording food. Quantification and analysis of the energy intake and nutrient composition were performed by the software MètaDieta ^®^ (Me.Te.Da S.r.l., San Benedetto del Tronto, Italy). GI and GI were calculated as previously described by Verduci et al. [8].

#### 2.2.2. Biochemistry 

Hematological and biochemical status was evaluated according to patients’ routine check-ups at both T0 (baseline) and T1 (6 months of GMP intake). Analyses included: complete blood count, fasting glucose, fasting insulin, LDL, HDL, total cholesterol, triglycerides, iron status (plasmatic iron, ferritin, transferrin), folate, vitamin B12, vitamin D (25OH), prealbumin, albumin, total proteins, phosphorus, calcium, alkaline phosphatase, and urea. We also collected all the Phe values of the dried blood spots (DBS) performed in the six months before the enrollment and during the six months of the study, calculating the medium value. The frequency of the DBS assay was dependent on age, with a minimum of 1 DBS/month in adulthood.

### 2.3. Short-Chain Fatty Acids Quantification

Fecal short-chain fatty acids (SCFAs) quantification was performed by gas chromatography as previously described [16]. Before the experiment, feces were weighed (200 mg), suspended in double-distilled water (1 mL), and homogenized by a bead beater. An aliquot corresponding to 60 mg of feces (300 µL) was acidified with phosphoric acid (pH < 2), diluted with water to 1 mL, and extracted twice with diethyl ether-heptane (2 × 500 µL, 1:1 *v/v*). The aqueous phase was frozen at −80 °C, and the organic layer was collected for the analysis with a gas chromatography GC-2010 coupled to a flame ionization detector (FID) (Shimadzu, Kyoto, Japan). The GC was equipped with a Stabilwax-DA fused-silica capillary column (30 m, 0.53 mm i.d. with a film thickness of 0.25 µm). The GC/FID conditions employed were injection volume 1 µL, split ratio 5, helium flow rate 4.4 mL/min, injection temperature 200 °C, and detector 250 °C. The initial column temperature was 80 °C and held 2 min, ramped to 250 at 8 °C/min, and kept at this temperature for 5 min with a total run time of 28 min. Quantification of the SCFAs was obtained through calibration curves of acetic, propionic, butyric, isobutyric, and isovaleric acids in concentrations between 0.25 and 10 mM. Areas of the analytes were normalized by the response of 2-ethylbutyric acid, used as the internal standard. SCFA concentrations were expressed as µmol/mg feces. 

### 2.4. Gut Microbiota Sequencing and Analysis 

Gut microbiota characterization was performed at the time-points T0 (baseline) and T1 (after 6 months of GMP intake), thus collecting two samples from each patient. Stools were kept at −80 °C until use.

Fecal DNA extraction was performed by the PSP Spin stool DNA kit (Invitek Molecular GmbH, Berlin, Germany) according to the manufacturer’s instructions and stored until use at −20 °C. The V3–V4 hypervariable regions of the bacterial 16S rRNA gene were amplified with a two-step barcoding approach according to the Illumina 16S Metagenomic Sequencing Library Preparation (Illumina, San Diego, CA, USA). Briefly, DNA samples were amplified with dual-index primers using a Nextera XT DNA Library Preparation Kit (Illumina), while library concentration and quantification were determined using a KAPA Library Quantification Kit (Kapa Biosystems, Woburn, MA, USA) and Agilent 2100 Bioanalyzer System (Agilent, Santa Clara, CA, USA), respectively. The libraries were pooled and sequenced with a MiSeq platform (Illumina) for 2 × 250 base paired-end reads, and a total of 2.5 Gbases raw reads were obtained.

The obtained 16S rRNA gene paired sequences were merged using Pandaseq (release 2.5; [17]). Reads were filtered by trimming stretches of 3 or more low-quality bases (quality < 3) and discarding the trimmed sequences whenever they were shorter than 75% of the original one. Bioinformatic analyses on gut microbiota were conducted using the QIIME pipeline (release 1.9.0; [18]), clustering filtered reads into Operational Taxonomic Unit (OTUs) at 97% identity level and discarding singletons as possible chimeras. 

Alpha-diversity was computed through the QIIME pipeline using the Chao1, the number of OTUs, Shannon diversity, and Faith’s Phylogenetic Diversity whole tree (PD whole tree) metrics. To compare the microbial community structure of the subjects, weighted and unweighted UniFrac distances were used. The phylogenetic assignment was performed via the RDP classifier [19] against the SILVA database (release 132; [20]), from phylum to genus level.

### 2.5. Statistical Analysis 

Clinical, nutritional, and taxonomic data differences between paired samples were established through the non-parametric Wilcoxon signed-rank test (R version 3.6.3, R Foundation for Statistical Computing, Vienna, Austria; used through RStudio, version 1.2.1335, RStudio, Inc., Boston, MA, USA). For the microbiota characterization, statistical evaluation among alpha-diversity indices was performed by a non-parametric Monte Carlo-based test within the QIIME pipeline, while beta-diversity differences were evaluated through the Permanova test (adonis function) in the R package vegan (version 2.0-10; [21]). Genus and SCFA correlation were assessed through the “gplots” R package (version 3.1.1), which evaluated statistical differences with the asymptotic *p*-value method. All *p*-values < 0.05 were considered significant.

## 3. Results

### 3.1. Cohort Description

Samples were collected from nine PKU patients (four adults and five pediatric subjects), at baseline and after 6 months of GMP intake, for a total of 18 total samples. Five patients had a classic (severe) PKU, while four patients had a mild PKU. The mean age of the enrolled subjects was 20 (age range 7–38), with a prevalence of males (7/9 subjects).

#### 3.1.1. Body Composition Analysis

The median value of body mass index (BMI) in the population was 18.1 kg/m^2^ (± 1.5), while the mean BMI z-score in the pediatric population was −0.2 (±1.7). All subjects were of normal weight according to the Centers for Disease Control and Prevention charts [22], except one pediatric subject who showed a BMI z-score indicative of underweight. FM was found increased in patients at T1 (median value 15.2% at T0 compared to 19.6% at T1; *p* = 0.004); total fat mass rose from 7.9 kg to 9.7 (*p* = 0.004), while we observed a reduction in the % of lean mass (84.8 to 80.4, *p* = 0.014) without a significant change in Fat-free Mass (FFM) in Kg (42.2 to 40.4, *p* = 0.652). FM % value in the pediatric population increased between T0 and T1 but remained within the reference values for age [23]. In detail, pediatric patients showed a median FM shifted from the 10th to the 25th percentile. For adults, female median FM% changed from the 25th to the 50th percentile [24], whereas male subject FM % did not change. Nonetheless, BMI remained overall stable during the study period.

#### 3.1.2. Dietary Assessment and Biochemical Analysis

The dietary intervention was aimed at replacing part of the protein substitute (PS), consisting of free amino acids, with GMP. Previously, none of the patients had ever used GMP before, and all of them were following a similar diet, characterized by natural protein restriction, use of Phe-free-l-amino acid supplements, and low-protein foods in order to reduce the intake of phenylalanine (Phe) according to their specific tolerance. None of the enrolled subjects had ever used tetrahydrobiopterin (BH4) treatment. About the protein substitutes, they were taking different brands of amino acid mixtures according to patient characteristics (age and weight) and taste preferences, but none of the substitutes was supplemented with prebiotics and/or probiotics. On the contrary, GMP supplementation was achieved by using products manufactured by the same company.

The protein replacement was completed in four out of nine patients (100%), while in the other five subjects, the percentage of GMP at T1 ranged from 33% to 79%. Nutritional intake values are reported in Appendix A and biochemical parameters in Table 1. Despite the energy intake, glycemic index, and soluble fibers being slightly increased, both glycemia and insulin plasmatic levels remained in the normal range. Total fiber intake did not change significantly during the study period, and no differences were seen according to the Phe tolerance.

The average Phe value from dried blood spot (DBS) showed a non-significant increase, remaining in all subjects below the target Phe levels for age, according to European guidelines for PKU management [25]. Indeed, in the 6 months prior to enrollment, patients < 12 years of age showed a mean Phe value of 211 µmol/L (±0.3), while in the study period, the mean was 302.8 µmol/L (±9.7). Similarly, in the population ≥ 12 years old, the mean Phe value before the dietary intervention with GMP was 378.3 µmol/L (±122.9), while in the study period, it was 387.9 µmol/L (±78.4). 

Some vitamin and mineral intakes showed significant differences before and after GMP introduction: intakes of calcium and vitamin B12 were found to be reduced (respectively, from 2038.4 mg to 1392 mg, *p* = 0.039 and from 10.7 µg to 5.4 µg, *p* = 0.008), without calcium and vitamin B12 plasmatic level alterations. Moreover, considering that PKU patients are generally supplemented with vitamin D, a significant increase in plasmatic vitamin D levels (from 32.2 ng/mL at T0 to 44.7 ng/mL at T1, *p* = 0.027) was found in our cohort, despite the unchanged vitamin D supplementation and oral intake. Furthermore, a significant reduction in alkaline phosphatase levels was observed (*p* = 0.027). 

#### 3.1.3. Short-Chain Fatty Acid Quantification

SCFAs amounts did not reveal any significant change during the intervention. Acetate levels slightly increased after 6 months of GMP intake, from a median of 25.8 (IQR 13.9) at T0 to 31.6 (IQR 34), while median butyrate concentration slightly decreased during the intervention (12.7, IQR 12.4 at T0; 9.8, IQR 9.9 at T1, *p* = 0.945). Moreover, we observed huge subject-dependent variations and discordant trends, with some patients showing an increase and others a decrease (Appendix A). The other fecal SCFA concentration analyzed (i.e., isobutyrate, isovalerate, and propionate) remained stable (Appendix A).

### 3.2. Microbiota Characterization

#### 3.2.1. Biodiversity Assessment

Alpha- and beta-diversity did not show significant differences after the GMP supplementation period (Figure 1A). Indeed, a bacterial biodiversity trend was not clearly observed. Similarly, GMP patients did not show a net dissimilarity over time in the beta-diversity either along the Unweighted or the Weighted Unifrac distances (Figure 1B,C).

#### 3.2.2. Taxonomic Comparisons

Bacterial groups analysis revealed a non-significant reduction in the Verrucomicrobia phylum from T0 to T1 (*p* = 0.054) and a trend toward an increase in the Firmicutes during the GMP intake (56.75% at T0 vs. 75.85% at T1, *p* = 0.641). Although not significant, at the family level (Figure 2A), *Ruminococcaceae* and *Bacteroidaceae* showed opposite trends over time, increasing and decreasing after GMP intake, respectively. *Ruminococcaceae* median abundance consistently rose from 22.88% at T0 to 40.22% at T1, while *Bacteroidaceae* decreased from 21.75% to 13.64%. At the genus level (Figure 2B), *Agathobacter* relative abundance increased from 0.23% at T0 to 2.61% at T1 (*p* = 0.008), and *Subdoligranulum* rise was even more noticeable, from 7.04% at T0 to 17.66%, although not uniformly among subjects. A group of the *Lachnospiraceae* family (“uncultured” genus) was found significantly reduced from T0 to T1 (1.08% to 0.49%, *p* = 0.023), as well as *Akkermansia*, although non statistically significant (2.26% vs. 0.01%, *p* = 0.055), while *Escherichia-Shigella* relative abundance was slightly increased (from 0.24% to 0.70%, *p* = 0.078). Relative abundances at the three phylogenetic levels are reported in Appendix A.

Microbiota analysis did not reveal any significant difference (Mann–Whitney U test, *p* > 0.05) in the main (>1%) relative abundances at the phylum, family, and genus level when comparing pediatric and adult patients over time. Similarly, no significant difference was detectable by grouping subjects according to the percentage of GMP reached at T1 (those who completed the protein replacement −100% GMP; four patients and those who did not—GMP ranging from 33% to 79%; five patients).

### 3.3. Correlations between Microbiota, SCFA, and Biochemical Parameters

Figure 3 shows the relationship between fecal short-chain fatty acid concentrations and bacterial taxa. “*Ruminococcus 2”* positively correlates with butyrate (R = 0.647), and *[Eubacterium] coprostanoligens* group with propionate (R = 0.427; *p* = 0.036). Four genera were characterized by positive correlations with all SCFAs: *Blautia*, *Dialister*, “*Ruminococcus 2”* , and *Akkermansia*. On the other hand, *Bacteroides, Parabacteroides*, and *Alistipes* were clustered together as they all showed negative relationships with butyrate (respectively, R= −0.566, R= −0.504, R= −0.696; *Alistipes*, *p* = 0.011), acetate, and propionate. *Agathobacter* showed a negative correlation with isovalerate (R= −0.547).

We further evaluated (Figure 4) possible correlations between the gut microbiota, butyrate, and the key clinical parameters (alkaline phosphatase, inorganic phosphorus, DBS Phe, and the plasmatic concentrations of vitamin B12, vitamin D, calcium, and Phe). Among taxa, we considered the most abundant and significant genera (*Bacteroides, Subdoligranulum, Faecalibacterium, Blautia, Agathobacter, uncultured_Lachnospiraceae*). Alkaline phosphatase, the uncultured members of the *Lachnospiraceae* family, and DBS Phe reported the overall highest magnitudes of separation. Vitamin D, calcium, and butyrate share separation loadings on the component 3 axis, with the former in a negative and opposite way to the others. PHE amounts, both plasmatic and DBS, reported a strong contribution along with the component 1 and toward the T1 group after the GMP intervention; similarly, *Subdoligranulum* and *Agathobacter* trends also indicate a bias toward the T1 group.

## 4. Discussion

Recent studies suggest a GMP influence on the gut microbiota in the in vitro and ex vivo models [14,26,27] by promoting the gut microbiota bacterial diversity. Experiments conducted on piglets showed a reduction in *Escherichia coli* attachment to the intestinal mucosa and an increase in the relative abundance of lactobacilli species [28]. In a rodent model with GMP supplementation from weaning (3 weeks of age) through young adulthood, GMP showed prebiotic properties along with a possible anti-inflammatory function. Specifically, a reduced *Desulfovibrio* relative abundance and an increase in SCFAs (acetate, propionate, and butyrate) production was observed in PKU mice on a GMP diet compared to PKU mice on low-Phe amino acids or a casein-rich diet [14]. 

On the other hand, in humans, a 4-week randomized study conducted on twenty-four healthy subjects in 2021 has found no differences in fecal microbiota composition between adults consuming GMP or placebo. Similarly, no differences were found in body weight, fecal calprotectin, and SCFA levels at baseline and at week 4 between the two groups [29].

Despite in our cohort, the replacement of the protein substitute resulted in a slight variation in macronutrient and micronutrient intakes, we confirmed that the GMP supplementation does not strongly affect microbiota composition, as both alpha- and beta-diversity did not show significant differences over time.

Previous studies described PKU gut microbiota as enriched in the genera *Blautia* and *Clostridium* [11,30] while depleted in two key butyrate-producing genera, *Faecalibacterium* and *Roseburia*, with a consequent reduction in the concentrations of total fecal SCFAs and butyrate [8,31]. 

The relative abundance analysis unveiled a significant increase in *Agathobacter* genus, formerly annotated as the *Eubacterium* genus. Interestingly, *Agathobacter* belongs to the *Lachnospiraceae* family and is a butyrate-producing microorganism, thus considered a beneficial commensal and potential probiotic candidate [32]. Furthermore, according to our results, Phe plasmatic levels negatively correlate with *Agathobacter* after a GMP intervention, suggesting possible beneficial effects of this genus on the Phe metabolism. These data are in agreement with recent findings that showed a lower abundance of *Agathobacter* in PKU patients with neurological abnormalities [33]. 

Although in a subject-dependent manner, *Subdoligranulum* relative abundance was primed by GMP supplementation and positively correlated with *Agathobacter*. *Subdoligranulum* belongs to the *Ruminococcaceae* family, is closely related to *Faecalibacterium,* and was found enriched by prebiotic supplementation [34]. Van Hul et al. demonstrated the positive correlation of *Subdoligranulum* spp. abundance with a healthy metabolic status and showed that despite it being a butyrate producer, the supplementation with *S. variabile* in mice does not increase the fecal SCFAs concentration as expected [35]. 

Our results on SCFA levels before and after GMP supplementation support previous research findings, as we did not observe changes in the mean abundance of the studied microbial metabolites. Nevertheless, the rapid butyrate uptake by enterocytes, their preferred energetic source [36], might mitigate a detectable increase related to changes in the abundance of both *Agathobacter* and *Subdoligranulum*. Furthermore, our correlation analysis showed that the relative abundance of genera negatively associated with butyrate concentrations (i.e., *Alistipes, Bacteroides,* and *Parabacteroides*) was decreased by GMP supplementation, although not in a statistically significant way for all genera.

Butyrate-producing taxa have been recently positively correlated with both 1,25(OH)2D levels and the vitamin D activation ratio [37]. It is interesting to note that we observed an increase in vitamin D plasmatic levels, despite PKU subjects not changing their intake during the intervention and that these levels were opposed to the butyrate and calcium ones. Butyrate has been demonstrated to promote the overexpression of vitamin D receptors that, in turn, elicit a cell response to 1,25(OH)D promoting intestinal homeostasis and immune functions [38,39]. In addition to this, potential beneficial effects of vitamin D on gut microbiota were described [40], and this might be confirmed by our correlation data between vitamin D, *Agathobacter,* and *Subdoligranulum*.

Considering that a bone-related disorder is associated with an increase in alkaline phosphatase activity in response to calcium or vitamin D deficiency, the increase in vitamin D plasmatic levels observed in our cohort, together with the decrease in alkaline phosphatase levels, suggests an overall amelioration of calcium-phosphate homeostasis as a consequence of the GMP intake [41,42]. According to the European Food Safety Authority (EFSA) guidelines, indeed, vitamin D serum levels in infants and children, evaluated as the concentration of 25(OH)D, should be at least >50 nmol/L (20 ng/mL) in order to avoid a possible deficiency [43]. Nevertheless, international and national scientific societies consider a minimum cut-off of sufficiency the value of 75 nmol/L (30 ng/mL) [44].

Osteopenia and osteoporosis were described in PKU as a consequence of an inadequate intake of micronutrients and vitamin D, an outcome also observed in classical PKU patients with normal alkaline phosphatase. Osteopenia pathophysiology seems to correlate with the grade of PKU severity, possible alterations of bone metabolism, and the quality of protein intake. Indeed, intact proteins show a different biological value, and the percentage of them versus Phe-free L-amino acid supplements seem to negatively correlate with the entity of mineral bone disease [25]. GMP represents a natural protein, and our data suggest a possible beneficial indirect effect on bone health. In agreement, a mice model of low-Phe GMP supplementation compared with Phe-free L-amino acid supplements showed an improvement in bone density in the GMP group [45]. Moreover, GMP formulations, compared with amino acid medical foods, are associated with a reduced urinary calcium excretion and reduced dietary acid load, possibly reducing the risk of skeletal fragility [46].

Within our study, the anthropometric analyses showed an increase in fat mass (% and in kg) and a decrease in the lean mass (%), despite unchanged BMI, during the intervention period. However, we observed that FFM expressed in kg remains stable, which means that the lean mass was preserved in the study period; our data, therefore, may reflect a trend in our population towards the 50th percentile according to percentiles of the general population.

In this regard, we cannot exclude that our data can be biased by the national lockdown due to the COVID-19 pandemic that forced a reduction in almost all the usual fitness activities. In support of this hypothesis, a recent study by Daly and colleagues [47], analyzing the long-term effect of GMP supplementation in 29 PKU children versus 19 supplemented with AA formula, did not observe changes in the percentage of body fat mass and total fat mass. Vice versa, they reported a trend toward an improved body composition in subjects taking 100% GMP as a protein source.

The percentage of GMP-derived protein in our enrolled subjects varied from 30 to 100%, independently from Phe levels, without evident differences in terms of microbiota composition. This may represent a major limitation of this study since the nutritional needs of these subjects are different, and the diet needs to be adjusted to this factor. Indeed, despite GMP-based formulations being commonly considered more palatable, not all PKU patients similarly accepted the switching from the conventional AA formulas to GMP due to its different taste, excessive volume, or the high caloric intake of liquid formulations [45]. Other limitations of this study are represented by the small number of participants and the absent homogenization of low-proteins foods, which may have an impact on microbiota modulation.

## 5. Conclusions

In conclusion, this study suggests a possible positive effect on the microbiota induced by GMP, with an increase in beneficial bacteria such as *Agathobacter* and, in a subject-dependent manner, *Subdoligranulum*. Further studies on larger cohorts are needed to corroborate our data and to define a microbial signature identifying PKU subjects who can take the maximum advantage of GMP supplementation. Another issue to be addressed in the future is whether GMP effects are age-dependent. Currently, GMP intake has been suggested as a valid nutritional therapy for PKU patients [48], even if none of the literature studies enrolling young children has assessed the GMP impact on the microbial community yet [49]. It is predictable that early intervention could have more significant effects on the gut microbiota when compared with those observed in an already developed and stable ecosystem.

## Figures and Tables

**Figure 1 nutrients-14-01883-f001:**
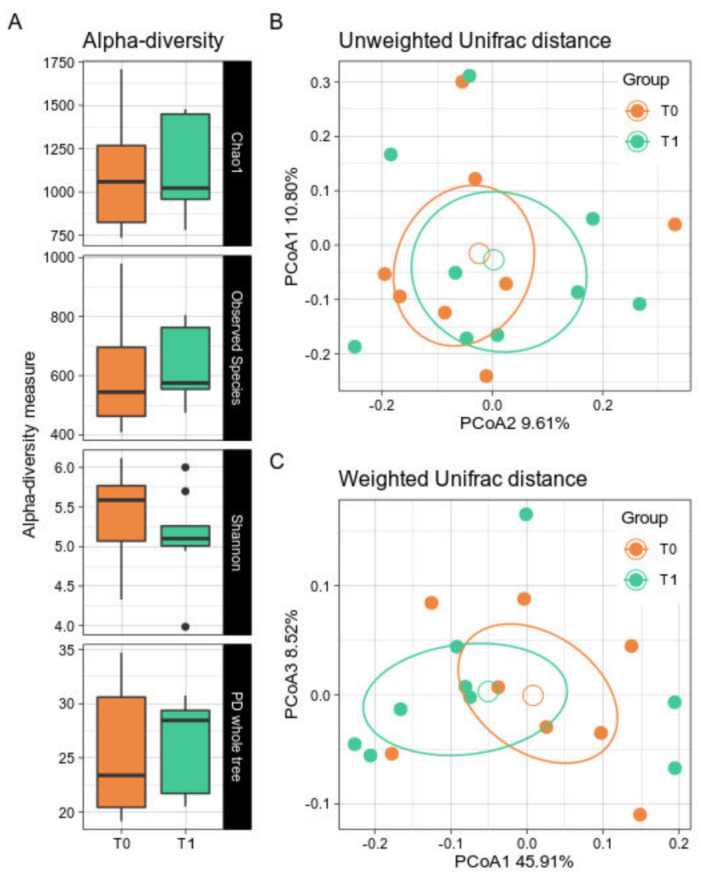
Alpha- and Beta-diversity between timepoints. PKU patients were sampled at baseline (T0) and after 6 months of GMP intake (T1). (**A**) Boxplots showing the metrics Chao1, Observed Species, Shannon index, and PD whole tree. Black dots represent outlier samples. (**B**,**C**). PCoA of the Unweighted and Weighted Unifrac distances with a boxplot of samples’ distribution for, respectively, the first and second coordinates and the first and third coordinates. T0 samples are depicted in orange; T1 in green. Solid dots represent single samples, empty circles represent the group average. For each group, confidence ellipse are reported. No statistically significant differences were observed among groups for both biodiversity indices.

**Figure 2 nutrients-14-01883-f002:**
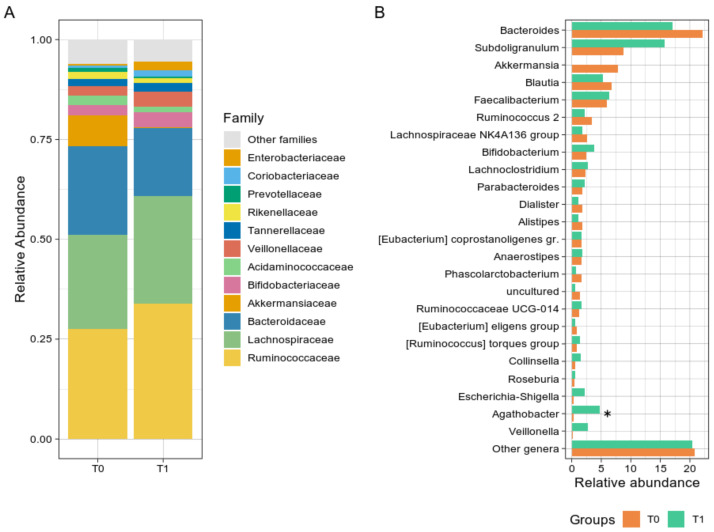
Taxonomy analysis. (**A**) Histogram chart showing the family level distribution of PKU patients at T0 and T1. (**B**) Bar plot of the main bacterial genera. For both panels, bacteria were selected according to their relative abundance (at least 1%); the rest of the abundances were grouped in the “Other genera” group. The asterisk indicates significant correlation (* *p* < 0.05).

**Figure 3 nutrients-14-01883-f003:**
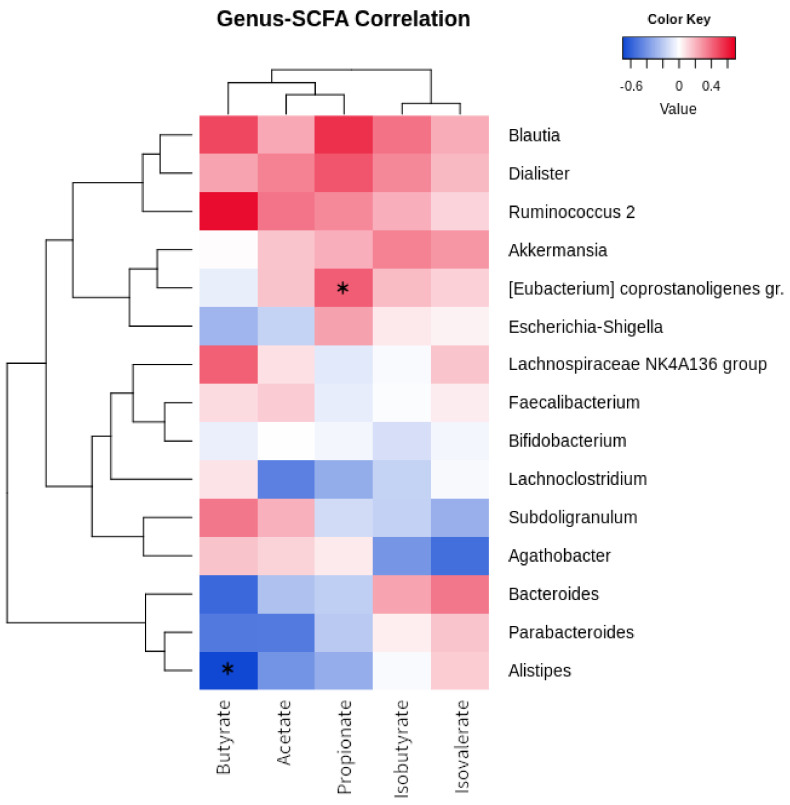
Correlation of bacterial genera and SCFA. Heatmap showing the Euclidean correlation R-values occurring between the most abundant bacterial genera and fecal SCFAs. Clustering was performed through a complete agglomeration method. Asterisk indicates significant (*p* < 0.05) correlation (asymptotic *p*-value).

**Figure 4 nutrients-14-01883-f004:**
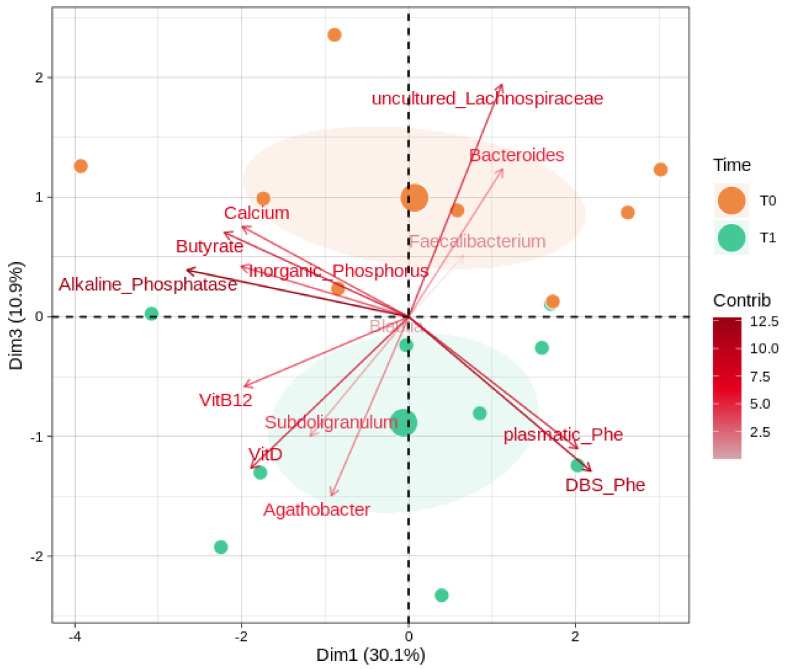
Microbiota, biochemical values, and SCFA interactions. PCA shows correlations between the most abundant genera and significantly changed ones (as reported in Appendix A), the key biochemical parameters, and butyrate. Samples are divided by timepoints: T0 (baseline) and T1 (after 6 months of GMP intervention). Color gradients and transparency depend on the variable loading contribution; the arrow length and distance from the middle indicate the magnitude of the separation. Positively correlated variables point to the same quadrant of the plot; negatively correlated variables point to opposite sides. The first and the third principal component (dimensions) are shown, along with their eigenvalues (percentage of variances).

**Table 1 nutrients-14-01883-t001:** Biochemical values at T0 and T1. Amounts are reported as median and interquartile range (IQR)**.** Statistical significance was determined through a non-parametric Wilcoxon paired signed-rank test; *p*-values below 0.05 were considered significant.

Variable	Median Value at T0; IQR (*n* = 9)	Median Value at T1; IQR (*n* = 9)	*p*-Value
**Protein nutritional status**			
Albumine (g/dL)	4.6; 0.3	4.5; 0.5	0.078
Transthyretin (prealbumin) (mg/dL)	24.8; 3.3	24.2; 5.1	0.426
Total protein (mg/dL)	7.5; 0.3	7.5; 0.8	0.068
**Vitamin and mineral status**			
Vitamin D (25-OH) (ng/mL)	32.2; 11.7	44.7; 21.9	0.027
Calcium (mg/dL)	9.9; 0.3	9.7; 0.4	0.138
Phosphorus (mg/dL)	4; 0.6	3.7; 0.5	0.024
Vitamin B12 (pg/mL)	854; 255	846; 74	0.359
Folate (ng/mL)	9.7; 7.2	15; 6.1	0.059
Alkaline phosphatase (U/L)	102; 115	72; 52	0.027
**Glucose metabolism**			
Glucose (mg/dL)	83; 11	86; 8	0.905
Insulin (µU/mL)	5.6; 4.1	5.1; 4.7	0.297
**Lipid profile**			
Total Cholesterol (mg/dL)	160; 14	163; 11	0.570
LDL Cholesterol (mg/dL)	96; 33	89; 26	0.820
HDL Cholesterol (mg/dL)	52; 16	54; 21	0.105
Triglycerides (mg/dL)	66; 41	89; 50	0.635
**Iron status**			
Plasmatic iron (µg/dL)	90; 38	117; 74	0.250
Ferritin (ng/mL)	45.2; 23.9	46.3; 8.9	0.652
Transferrin (mg/dL)	258; 42.0	243; 32	0.075
**Metabolic control**			
Blood Phe values (µmol/L) ^a^	300; 222.6	391.8; 94.8	0.496

^a^ median value of Phe levels assessed by DBS in the 6 months before the dietary intervention and the during study.

## Data Availability

Raw reads are available in NCBI Short Read Archive (SRA, http://www.ncbi.nlm.nih.gov/sra (accessed on 30 January 2022) under accession number PRJNA784527.

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
