# Peer review of "Glycomacropeptide Safety and Its Effect on Gut Microbiota in Patients with Phenylketonuria: A Pilot Study"

_nutrients, 2022, doi:10.3390/nu14091883_

Round 1

Reviewer 1 Report

This manuscript aims to study the effect of GMP supplementation on gut microbial community of PKU patients. The methods are clearly explained and the results are properly described. Nevertheless, I have a major concern regarding this study due to the recruited population. The results are well described and properly discussed.  From the studied population I find it hard to conclude with certainty.

The thematic and the problem of this paper are very pertinent but it is very specific for this population, PKU patients, under this specific diet.

The diet is very restrictive for these patients and microbiota is highly influenced by this. The studied population are 9 patients with PKU, 4 adult and 5 pediatric with ages (that include children and adolescents) from 7-38 years and mostly male. It is not described if any of them are using BH4 or having the same diet (same type of products).

The nutritional needs of these subjects are different and the diet is certainly adjust to this factor. So the GMP intake can be variable between individual but that is not clear in the paper. Also, not all of them change to 100% GMP, some stayed with 30% GMP. Given this, and the small number of participants of the study I would say this is a major issue for the conclusion of this paper.

The dietary assessment was made from food diary and nutritional values presented as supplementary material. Given that the population is so different (adults and pediatric) the amount and type of food other than protein substitutes can be very different in what it is prebiotic and phytochemical composition with impact in microbiota modulation. Was this variable controlled?

GI and GL was calculated based on the 3-day food diary. GI is characteristic of food, it is not understandable in the paper the purpose of this determination.  

The author state that there were huge subject-dependent variations and discordance trend in short chain fatty acid determination. Couldn’t this be a consequence of a very diverse population and still with some dietary differences (100% GMP to 30%). The vertical lines in supplementary figure 2 doesn’t help to see the variations. The figure could be clear.

It could be helpful to stratify the microbiota diversity by age or by % of GMP. The n is going to be smaller but the microbiota composition from an adult, a children or an adolescent can be significantly different or the GMP dose could also be influencing bacterial modulation.

The Agathobacter and Subdoligranulunm were detected in all samples? Since you have a small number of participants it could be helpful to understand the frequency of these genera and if the effect is consistent among individuals.

Author Response

Comment # 1: This manuscript aims to study the effect of GMP supplementation on gut microbial community of PKU patients. The methods are clearly explained, and the results are properly described. Nevertheless, I have a major concern regarding this study due to the recruited population. The results are well described and properly discussed. From the studied population I find it hard to conclude with certainty. The thematic and the problem of this paper are very pertinent, but it is very specific for this population, PKU patients, under this specific diet.

The diet is very restrictive for these patients and microbiota is highly influenced by this. The studied population are 9 patients with PKU, 4 adult and 5 pediatric with ages (that include children and adolescents) from 7-38 years and mostly male. It is not described if any of them are using BH4 or having the same diet (same type of products).

  • We thank the Reviewer for the feedback and the positive comments. We agree that such a special diet could strongly impact PKU gut microbiota. To address this concern, we have widely reworked the results section, better describing the inclusion and exclusion criteria, the diet of our population and their characteristics (lines 205-213).

All the patients were following a similar diet characterized by natural protein restriction, use of Phe-free-l-amino acid supplements and low-protein foods, in order to reduce the intake of Phe according to their specific tolerance. Concerning protein substitutes, despite differences in the formulas linked to age, weight and taste preferences, none was supplemented with either probiotics or prebiotics. With regard to low-protein foods, their intake was not modified during the dietary intervention, and all the subjects freely used the preferred ones.

None of the subjects was taking BH4 before and during the study.

Comment # 2: The nutritional needs of these subjects are different, and the diet is certainly adjust to this factor. So the GMP intake can be variable between individual but that is not clear in the paper. Also, not all of them change to 100% GMP, some stayed with 30% GMP. Given this, and the small number of participants of the study I would say this is a major issue for the conclusion of this paper.

  • We agree with the reviewer’s observation and we underlined this limitation in the discussion. Nevertheless, following the reviewer suggestion, we performed gut microbiota analysis after grouping subjects according to the reached percentage of GMP (100%, and 30-79%) to test whether significant changes could rely on the GMP % or whether a dose-dependent effect was visible. Although no significant differences were observed, we added this data in the results section (lines 280-285).

Comment # 3: The dietary assessment was made from food diary and nutritional values presented as supplementary material. Given that the population is so different (adults and pediatric) the amount and type of food other than protein substitutes can be very different in what it is prebiotic and phytochemical composition with impact in microbiota modulation. Was this variable controlled?

  • We have included more information about the diet of our patients (lines 205-213). Indeed, inter-individual differences according to age and Phe tolerance were present, but the dietary pattern was the same and all patients were taking amino acid mixtures with low protein products. None of the formula or the products was supplemented with probiotic and/or prebiotics.

Comment # 4: GI and GL were calculated based on the 3-day food diary. GI is characteristic of food, it is not understandable in the paper the purpose of this determination.

  • We have reported GI and GL since in a previous paper (Verduci et al., 2018- doi:10.1016/j.numecd.2018.01.004) the two indices resulted increased in PKU patients following the special diet compared with subjects with mild hyperphenylalaninemia on free diet. The quality of foods seems to be the main reason of the increased GI e GL. Indeed, special low protein foods are added with glucose, dextrose or sugar to ameliorate the palatability. We have added a sentence in the introduction explaining the rational for GI and GL calculation (lines 60-61).

Comment # 5: The author states that there were huge subject-dependent variations and discordance trend in short chain fatty acid determination. Couldn’t this be a consequence of a very diverse population and still with some dietary differences (100% GMP to 30%).

  • We agree with the Reviewer for the insightful and constructive feedback. We run new bioinformatics analyses to verify whether GMP percentage could drive the observed subject-dependent variations. As better specified in response to comment #7, we did not find statistically any difference according to GMP intake.

Comment # 6: The vertical lines in supplementary figure 2 doesn’t help to see the variations. The figure could be clear.

  • We believe the Reviewer is referring to Supplementary Figure 1. The vertical lines (Upper and Lower Whiskers) are part of the boxplot and indicate value variability outside the upper and lower quartiles. To make the figure clearer, we amended it by differentiating the whiskers from the lines connecting the samples.

Comment # 7: It could be helpful to stratify the microbiota diversity by age or by % of GMP. The n is going to be smaller but the microbiota composition from an adult, a children or an adolescent can be significantly different or the GMP dose could also be influencing bacterial modulation

  • We thank the Reviewer for the suggestion. Indeed, we have performed a statistical analysis (Mann-Whitney U test) to see microbiota differences dividing by Pediatric/Adult age over time (T0 vs T1), and none of the bacterial groups with a relative abundance >1% were detected. Minor differences were seen for a few genera with scarce relative abundances (avg <0.1%), such as Fusicatenibacter (Pediatric median relative abundance 0.061 at T0 and 0.053 at T1 vs 0.12 and 0.71 in adults; p=0.036).

As per the GMP percentage, when grouping patients in those who completed the protein replacement (100% GMP; 4 patients) and those who did not (GMP ranging from 33% to 79%; 5 patients) we did not find statistically any difference between groups, probably because of the small numbers (4 vs 5).

We added these additional results in the proper section in the manuscript (lines 280-285).

Comment # 8: The Agathobacter and Subdoligranulunm were detected in all samples? Since you have a small number of participants it could be helpful to understand the frequency of these genera and if the effect is consistent among individuals.

  • Yes, they were both detected in all samples, with relative abundances ranging (min-max values) 0.02%-37.11% for Subdoligranulum, and 0.005%-19.90% for Agathobacter; average values divided by time-point were 8.78% (T0) - 17.32% (T1) for Subdoligranulum, and 0.36% (T0) - 4.75% (T1) for Agathobacter.

Reviewer 2 Report

I would like to see a little more characterisation of the 9 PKU subjects in the "Cohort Description" section:

  1. Can you state explicitly that all 9 subjects were GMP naive - ie had never had it before and had only ever been treated by amino acids?
  2. Mean phe levels are given - are these lifetime mean phe levels?  Of the 5 adult subjects, oldest = 38 and more subjects were male, had any ever ceased dietary treatment for PKU, is this known?
  3. What was the phe tolerance of the subjects? ie did all 9 subjects have classical PKU, mild PKU etc?  This could influence dietary prebiotic intake eg cereal grains etc
  4. Would be useful to see reference to the intake of dietary fibres - the nutritional intake data is not much commented on and there were some variations between subjects' microbiota at T1 - it's possible that this could be linked to age (you allude to this in the conclusion) or maybe diet.  (You say this has not changed, but it would be interesting to see some basic values eg on fibre/non-starch polysaccharide intake).

Study limitations - in either discussion or conclusion

  1. The discussion is really well written.  I would value a comment on the duration of this study, is this a limitation to the findings?  Why was 6 months chosen as the end point and would a later/2nd end point capture any other data?  How quickly do gut microbiota change in response to dietary changes - would all changes in microbiota normally have happened by 6 months? I think the answer is yes all changes would probably have happened by T1, as you do not mention study duration at all.... it would be helpful to readers to say this?

A great paper, very interesting, especially about calcium metabolism but also the very interesting findings discussed in lines 317-324. Thank you.

Author Response

REVIEWER 2

Comment # 1: Can you state explicitly that all 9 subjects were GMP naive - ie had never had it before and had only ever been treated by amino acids?

  • We thank the reviewer for this valued comment. Indeed, all the patients were GMP naive since previous GMP intake was one of the exclusion criteria (that was missing in the M&M section, now amended: line 90)

Comment # 2: Mean phe levels are given - are these lifetime mean phe levels? Of the 5 adult subjects, oldest = 38 and more subjects were male, had any ever ceased dietary treatment for PKU, is this known?

  • We thank the Reviewer for this suggestion. We have specified that the average values refer to mean Phe values in the 6 months before the enrollment and of values during the six months of the study. According to their age, patients performed at least 1 DBS/month and none of them ever skipped follow-up visits. Moreover, none of the patients ever discontinued dietary treatment or was lost at follow-up in her/his life.

Comment # 3: What was the phe tolerance of the subjects? ie did all 9 subjects have classical PKU, mild PKU etc? This could influence dietary prebiotic intake eg cereal grains etc

Each patient displayed a specific tolerance to Phe and thus the granted natural proteins intake varied. Besides, the study did not include patients with extremely mild forms, consequently macronutrients from natural proteins were minimal compared to overall protein intake. Furthermore, we did not find significant variations between T0 and T1 in the intake of macronutrients. We added a sentence to better describe the cohort characteristics in relation to Phe tolerance (lines 184-185).

Comment # 4: Would be useful to see reference to the intake of dietary fibers - the nutritional intake data is not much commented on and there were some variations between subjects' microbiota at T1 - it's possible that this could be linked to age (you allude to this in the conclusion) or maybe diet. (You say this has not changed, but it would be interesting to see some basic values eg on fibre/non-starch polysaccharide intake).

  • We thank the Reviewer for pointing out this aspect. Due to the relevance of fibers in shaping the gut microbiota, we specify in the text that fiber intakes did not change significantly during the study period (lines 218-220).

Comment # 5: The discussion is really well written. I would value a comment on the duration of this study, is this a limitation to the findings? Why was 6 months chosen as the end point and would a later/2nd end point capture any other data. How quickly do gut microbiota change in response to dietary changes - would all changes in microbiota normally have happened by 6 months? I think the answer is yes all changes would probably have happened by T1, as you do not mention study duration at all.... it would be helpful to readers to say this?

  • Gut microbiota responds very quickly to dietary changes, but short-term intervention usually results in transient variations and the microbiota of the participants return to baseline within a few days after the discontinuation. On the contrary, long-term intervention (i.e., > 6 months) allows for defining stable changes in the microbial community. We added this aspect in the text (at the end of the introduction section) Moreover, the strengths and limitations of the study have been detailed.

Round 2

Reviewer 1 Report

This manuscript has improved with the revision, it became clear and understandable. I still think this is a very small and heterogenous population so the number of participants should be higher in order to increase confidence of the conclusions.